# Association between obesity and neurodevelopmental delay risk in children under five years: A study from Tumbes, Peru

Miriam Arredondo-Nontol[1,2], Rodolfo Arredondo-Nontol[1,2], Narcisa Reto[1,3*],
Alexis Germán Murillo Carrasco[4,5]

1 Hospital Carlos Alberto Cortez Jimenez Essalud Tumbes, Tumbes, Peru, 2 Universidad Nacional
de Tumbes, Tumbes, Peru, 3 Universidad Cesar Vallejo, Piura, Peru, 4 Instituto de Evaluación
de Tecnologías en Salud e Investigación - IETSI, Seguro Social de Salud - EsSalud, Lima, Peru,
5 Organization for Medical Innovation and Collaboration for Sciences—OMICS, Lima, Peru

* nereto@hotmail.com

## Abstract

### Background

Childhood obesity is an emerging public health concern in low- and middle-income countries and may be associated with early neurodevelopmental vulnerability. Evidence on this association during early childhood remains limited, particularly in Latin American settings.

### Objective

To evaluate the association between childhood obesity and neurodevelopmental delay risk in children under five years of age attending public healthcare facilities in Tumbes, Peru, and to develop a multivariable nomogram for probabilistic risk estimation.

### Methods

An analytical cross-sectional study was conducted between 2022 and 2024 among children aged 0–59 months receiving care at two EsSalud healthcare facilities in Tumbes. Neurodevelopment was assessed using the Evaluación del Desarrollo Infantil (EDI), classifying children as having normal development, developmental lag, or being at risk of developmental delay. Childhood obesity was defined using WHO weight-for-height standards. Sociodemographic, clinical, and behavioral variables were collected. Associations were evaluated using proportional odds ordinal logistic regression guided by a directed acyclic graph. A nomogram was developed based on the final model and internally validated using bootstrap resampling (1,000 iterations).

**Data availability statement:** An interactive calculator-like interface was developed using ShinyApps (RStudio®), which allows the child's individual characteristics to be entered and the estimated probability of belonging to each developmental category obtained. The resulting web application is available for open access at: https://omicsperulab.shinyapps.io/NeuroDev_Calculator/, the minimal dataset and codes used could be revised in https://github.com/Murillo22/Obesity_Peru_2025, and the full Markdown-generated HTML format can be reviewed at https://murillo22.github.io/Obesity_Peru_2025/Code.html. The full report, prepared in accordance with the STROBE (Strengthening the Reporting of Observational Studies in Epidemiology) guidelines [43], is available in Supplementary Material 3 and provides additional details on the multivariable modeling approach and nomogram development.

**Funding:** This study was partially funded by the Institute for Health Technology Assessment and Research (IETSI), ESSALUD, through its Mentoring Program, awarded by Miriam Arredondo-Nontol, Rodolfo Arredondo-Nontol, and Narcisa Reto. No additional external funding was received for this study. The funders had no role in study design, data collection and analysis, decision to publish, or preparation of the manuscript.

**Competing interests:** The authors have declared that no competing interests exist.

## Results

The final analytical sample included 431 children; 27% were classified as obese and 19% had anemia. According to the EDI, 58% had normal development, 36% developmental lag, and 6% were at risk of developmental delay. Childhood obesity was independently associated with higher cumulative odds of neurodevelopmental delay risk (OR = 2.73; 95% CI: 1.66–4.51). Male sex and older age group were also associated with increased risk, while higher caregiver knowledge of complementary feeding showed a protective association. Physical activity compliance and anemia were not independently associated in the multivariable model. The nomogram demonstrated acceptable internal discrimination (AUC > 0.7).

## Conclusions

Childhood obesity was associated with increased neurodevelopmental delay risk in children under five years of age. An explanation-informed nomogram using routinely available variables may support early risk stratification in primary care, although external validation is required before broader implementation.

## Introduction

Childhood obesity has become a global public health concern, with a steadily increasing prevalence in recent decades. The number of children under five affected by overweight or obesity rose from 32 million in 1990–41 million in 2016, with low-income countries being the most impacted [1,2]. In these settings, the prevalence of obesity among preschool-aged children was increased by 12.1% since 2000 [3]. In Peru, obesity rates among individuals under 19 years old have been reported to reach up to 25%, while among children under five, the rate is approximately 2.75%, a pattern largely attributed to global urbanization and increased access to high-calorie foods [4].

Childhood obesity is associated with serious health problems that often persist into adulthood [5]. For example, obese children experience more frequent episodes of headache, otitis, and musculoskeletal disorders [6]. Additionally, metabolic syndrome, hypertension, and dyslipidemia are increasingly common in obese children, and these conditions have been linked to insulin resistance and excess body fat [7].

Obesity may also negatively affect a child's psychosocial development and educational outcomes. Attention deficit hyperactivity disorder (ADHD), conduct disorders, depression, and learning disabilities occur more frequently among obese children [6,8]. Some authors suggest that the neurocognitive and behavioral impact of obesity may vary by age. During adolescence, it has been linked to poor academic achievement, memory impairment, and low self-esteem, which can hinder social development and limit professional opportunities due to impaired cognitive function [9–11]. In school-aged children, obesity has been associated with impaired gross motor skills, reduced attention span, and lower performance in mathematics and reading [12].

The precise age at which obesity begins to affect neurodevelopment remains uncertain, and existing studies on this topic are limited [13–15]. Some reports suggest that infants with overweight or obesity exhibit greater delays in motor and mental development compared to those with normal weight [11,14,15]. Motor test scores are often lower in children with higher body mass index (BMI), likely because excess weight hinders movement, requiring more physical effort to perform tasks [13].

A critical aspect of childhood obesity is its strong association with poor dietary habits and sedentary lifestyles [16]. High-calorie diets have been shown to impair cognitive function by damaging neuronal synaptic connections and affecting memory processes [17,18]. Furthermore, inadequate diets often lack essential nutrients, including those obtained from breastfeeding and key macro- and micronutrients such as proteins, omega-3 fatty acids, folates, and especially iron. Iron deficiency has been strongly linked to suboptimal neuronal development [17]. In early childhood, iron deficiency or anemia has been associated with cognitive deficits and delayed psychomotor and language development [19–21].

Another factor linked to childhood obesity is gut microbiota imbalance (dysbiosis), characterized by excessive populations of *Bacteroides* and *Lactobacillus* and reduced *Bifidobacterium* species during early childhood, compared to children with normal weight. This dysbiosis has also been associated with neurodevelopmental disorders and autism spectrum disorders. Gut microbiota development is positively influenced by vaginal delivery, exclusive breastfeeding, a healthy diet, and the avoidance of inappropriate use of antibiotics, steroids, and proton pump inhibitors [22].

Regarding physical activity, regular aerobic exercise has been shown to improve visual attention, processing speed, and performance in math and reading. Conversely, excessive screen time and sedentary behavior are associated with reduced academic performance, particularly in mathematics [18].

While there is evidence on the role of obesity and related factors (such as diet and physical activity) in early childhood neurodevelopment, understanding of this association remains limited, especially among children in low- and middle-income countries. In Peru, chronic malnutrition has historically been a public health problem [23], with its prevalence declining from 31.6% to 19.6% over the past five years; however, it has not yet been eradicated [24]. Paradoxically, childhood obesity has begun to emerge as a new epidemic, particularly on the northern coast, where a prevalence of 6.8% has been reported in children under five years of age [25].

Neurodevelopmental disorders constitute a multifactorial public health issue that requires early identification strategies. Multivariable predictive models offer a promising tool, especially when they integrate clinical, sociodemographic, and behavioral variables. Some studies have demonstrated the usefulness of nomograms in this field. For example, the risk of autism spectrum disorder has been predicted in Chinese children using immunological biomarkers (such as γδT cells) and dietary and behavioral variables, achieving an AUC of 0.905 (26 Zhang Risk prediction). Similarly, a predictive model for ADHD in children has been developed and externally validated, based on individual, family, and social factors, with excellent discriminative capacity (AUC=0.887 in the training cohort and 0.862 in the validation cohort) [26].

Despite these advances, the application of predictive models in Latin American contexts remains scarce. Therefore, this study aimed to evaluate the association between obesity and neurodevelopmental delay risk in children under five years of age in Tumbes, Peru, considering modifiable behavioral and clinical factors such as diet quality, physical activity, and anemia. Additionally, a multivariable predictive model was developed to estimate individual neurodevelopmental delay risk, and a clinical nomogram was constructed to support risk stratification and decision-making in primary care settings.

## Methods

### Study design and population

This was a cross-sectional analytical study conducted between May 2022 and November 2024 in two public healthcare facilities of the Peruvian Social Health Insurance (EsSalud), located in the city of Tumbes, northern Peru: the Zarumilla Health Center and the Carlos Alberto Cortez Jiménez Hospital.

The target population consisted of children under five years of age attending growth and development units and pediatric outpatient clinics at the participating facilities. Based on institutional records, the estimated source population during the study period was approximately 3,582 children.

A minimum sample size was estimated during the study planning phase based on the expected prevalence of neurodevelopmental delay risk. However, the final sample size was determined by consecutive recruitment of all eligible children who attended the participating facilities during the study period. Specifically, all children aged 0–59 months who met the eligibility criteria and whose parents or legal guardians provided written informed consent were invited to participate.

A total of 445 children were initially recruited. After exclusion of two ineligible records and one duplicate record, 442 eligible participants were included in the study population. Of these, 11 children (2.5%) had incomplete data in at least one key variable and were excluded from the analytical sample. The final analytical sample therefore comprised 431 children with complete data, as detailed in the Statistical Analysis section.

The focus on children under five years of age reflects the importance of early childhood as a critical period for brain development, during which neurodevelopmental trajectories are particularly sensitive to nutritional, metabolic, and behavioral factors. In addition, this age range corresponds to the validated target population for the Evaluación del Desarrollo Infantil (EDI), a standardized developmental screening tool designed to assess developmental status in children from birth to 59 months in primary care settings [27,28].

### Inclusion and exclusion criteria

**Inclusion criteria.** Children under five years of age attending the growth and development units or pediatric outpatient clinics at Zarumilla Health Center and Carlos Alberto Cortez Jiménez Hospital (EsSalud) during the study period, whose parents or legal guardians provided written informed consent to participate.

**Exclusion criteria.** Children with a previously diagnosed developmental delay, genetic syndromes (e.g., Down syndrome, Patau syndrome, Pierre Robin sequence), chronic debilitating illnesses (e.g., congenital heart disease, myopathies), severe malnutrition, epilepsy, or attention-deficit/hyperactivity disorder (ADHD) and Autism Spectrum Disorder (ASD).

### Ethics approval

The study was approved by the Institutional Research Ethics Committee of the Carlos Alberto Cortez Jiménez Hospital (EsSalud Tumbes) in August 2021 (Approval No. 002/2021 CEI-RATU-ESSALUD). Ethical approval was subsequently renewed on an annual basis during the data collection period, with approvals granted in 2022 (Approval No. 003/2022), 2023 (Approval No. 003/2023), and 2024 (Approval No. 005/2024), covering the entire duration of participant recruitment.

All procedures were conducted in accordance with the Declaration of Helsinki and national regulations governing research involving human subjects.

Written informed consent was obtained from the parents or legal guardians of all participating children prior to enrollment (Supplementary Material 1). Before consent, caregivers received a clear explanation of the study objectives, procedures (including interviews, clinical assessments, and blood sampling), potential risks and benefits, and the voluntary nature of participation, including the right to withdraw at any time without consequences. Consent was obtained in a language fully understood by the responsible adult, who was given adequate time to ask questions and make an informed decision. Given the age of the participants, verbal assent was not required.

### Assessment of childhood obesity

The primary variables of interest were childhood obesity and neurodevelopmental delay risk. Childhood obesity was assessed using the World Health Organization (WHO) growth standards, based on weight-for-height measurements

expressed as z-scores [29].Obesity was defined as a weight-for-height value above +2 standard deviations, corresponding to the >97th percentile for age and sex [30].

### Assessment of neurodevelopmental delay risk

Neurodevelopmental delay risk was assessed using the *Evaluación del Desarrollo Infantil* (EDI), a standardized developmental **screening tool** that evaluates fine-adaptive motor skills, gross motor skills, reflexes, hearing–language abilities, and personal–social development. The EDI classifies children into three categories: normal development, developmental lag, and risk of developmental delay. This instrument has been validated in multiple settings and has demonstrated adequate sensitivity and specificity for early detection of developmental alterations specificity [31,32].

### Physical activity compliance

Physical activity was assessed using a structured caregiver-reported questionnaire developed for this study, based on the World Health Organization (WHO) guidelines for physical activity in children under five years of age [33]. The instrument classified children according to age-specific recommendations and was administered to the child's primary caregiver during the clinical visit.

WHO physical activity recommendations were operationalized using age-specific criteria. For infants younger than one year, compliance was defined as engagement in active play several times per day, including supervised prone positioning ("tummy time") for a cumulative duration of at least 30 minutes while awake. For children aged 1–2 years, compliance was defined as participation in at least 180 minutes of physical activity of any intensity throughout the day. For children aged 3–4 years, compliance required a minimum of 120 minutes of total physical activity per day, including at least 60 minutes of moderate-to-vigorous physical activity, in accordance with WHO recommendations.

Based on caregiver responses, physical activity was classified dichotomously as "meets recommendations" or "does not meet recommendations" and was included as a categorical exposure variable in the multivariable analysis. Although this questionnaire was not a formally validated instrument, it was directly derived from international WHO guidelines and was designed to capture adherence to recommended physical activity thresholds in early childhood, a context in which continuous quantification of physical activity is challenging and caregiver-reported measures are commonly used [34].

### Anemia assessment

Anemia status was determined based on laboratory measurements. Hemoglobin concentration was obtained from venous blood samples collected as part of routine laboratory testing and analyzed at the institutional clinical laboratory. Anemia was defined using age-specific hemoglobin cut-off values established by the Peruvian Ministry of Health (MINSA) Clinical Practice Guideline for the prevention, diagnosis, and treatment of anemia [35].

For infants younger than 2 months, anemia was defined as hemoglobin concentration <13.5 g/dL, and for infants aged 2–5 months, as hemoglobin <9.5 g/dL. For children aged 6–59 months, anemia was defined as hemoglobin concentration <11.0 g/dL.

For the purposes of the present analysis, anemia was operationalized as a dichotomous variable (presence vs. absence of anemia) across all age groups, including children aged 6 months and older, to ensure analytic consistency and comparability throughout the study population. Although the MINSA guideline allows classification of anemia severity in children aged ≥6 months, severity categories were not used in this study

Because the study was conducted in Tumbes, a coastal region located near sea level, no altitude adjustment of hemoglobin values was applied.

## Maternal knowledge of complementary feeding

Maternal knowledge of complementary feeding was assessed using a structured questionnaire adapted from a previously validated instrument for use in Latin American populations [36]. The final version of the instrument consisted of 10 multiple-choice items, each with one correct answer, several incorrect options, and a "does not know" option.

Each correct response was assigned one point, while incorrect or "does not know" responses were assigned zero points, resulting in a total score ranging from 0 to 10, with higher scores indicating greater knowledge of complementary feeding practices.

For analytical purposes, total scores were categorized into three levels of maternal knowledge using predefined cutoff points: poor knowledge (0–3 points), fair knowledge (4–6 points), and good knowledge (7–10 points). This categorization was applied to facilitate interpretability and comparability with previous studies employing similar scoring approaches and validated instruments [37,38].

## Sociodemographic variables

Sociodemographic variables collected included the child's age (measured in months) and sex, as well as maternal age (measured in years) and maternal educational level. Maternal education was recorded as the highest level of formal education completed and categorized according to national education levels.

All sociodemographic variables were obtained through a structured caregiver interview at the time of enrollment.

## Data collection

Data were gathered through clinical observation during the EDI assessment and structured questionnaires administered to mothers. The researchers responsible for somatometry and survey performance, as well as the EDI assessment, worked independently to avoid information bias, and identification codes known only to the principal investigator were used. Information was entered into a Microsoft Excel 2016® database. Data cleaning was performed to identify and correct errors, omissions, and inconsistencies. Once data integrity was verified, categorical variables were coded for statistical analysis.

## Statistical analysis

The primary analytical objective of this study was explanatory. Accordingly, the statistical analysis was designed to estimate the association between childhood obesity and the risk of neurodevelopmental delay. Covariate selection was guided by a causal framework using a Directed Acyclic Graph (DAG) to appropriately control for confounding and avoid overadjustment [39].

All statistical analyses were conducted using R software (version 4.5.0). Descriptive statistics were summarized as absolute and relative frequencies (percentages) for categorical variables and as means with standard deviations for continuous variables. Comparisons across EDI developmental categories were performed using Pearson's chi-squared test or Fisher's exact test for categorical variables, and the Kruskal–Wallis test for continuous variables, as appropriate.

To evaluate factors associated with neurodevelopmental delay risk, an ordinal logistic regression model based on the proportional odds assumption was fitted [40]. Child development was categorized according to the EDI test as follows: EDI 1 (normal development), EDI 2 (developmental lag), and EDI 3 (risk of developmental delay). This approach allows estimation of the cumulative odds of belonging to a higher (i.e., more impaired) developmental category.

Independent variables included child's sex, age group, obesity status, anemia status, compliance with recommended physical activity, maternal knowledge of complementary feeding, maternal age, and maternal educational level. Results are presented as odds ratios (ORs) with 95% confidence intervals (95% CI) and corresponding p-values. ORs greater than 1 indicate increased cumulative odds of being classified in a worse developmental category.

The multivariable analysis was conducted using complete cases only (N = 431). Statistical significance was defined as a two-sided p-value < 0.05.

Missing data were assessed prior to analysis. Of the 445 children initially recorded, three were excluded before analysis due to ineligibility (n = 2) or duplicate records (n = 1). Among the remaining participants, 11 children (2.5%) had incomplete data for at least one key variable, including missing hemoglobin measurements (n = 10) or missing maternal questionnaire data (n = 1). Given the low proportion of missing data and the absence of evidence suggesting systematic missingness, analyses were conducted using a complete-case approach, resulting in a final analytical sample of 431 children (96.9%). No data imputation procedures were applied.

### Directed Acyclic Graph (DAG)

To ensure appropriate control for confounding in the multivariable analysis, a causal inference framework was applied through the construction of a Directed Acyclic Graph (DAG) [41]. This graphical model allowed the representation of hypothesized relationships between the main exposure (childhood obesity) and the outcome (neurodevelopmental delay risk), incorporating additional variables based on the scientific literature.

The DAG was created using the DAGitty.net platform and enabled identification of the minimal sufficient adjustment set to block backdoor paths, without adjusting for mediators or colliders. Based on the DAG analysis, the following variables were selected for adjustment: maternal age, maternal education level, maternal knowledge of complementary feeding, physical activity compliance, and anemia. The full DAG, along with causal assumptions and source references, is presented in Supplementary Material 2.

### Model training and validation

To assess the robustness and internal consistency of the ordinal logistic regression model, a resampling approach with 1,000 repetitions was implemented. In each iteration, the dataset was randomly split into a training set (80%) and a validation set (20%) using stratified sampling to preserve the distribution of the ordinal outcome variable, defined as neurodevelopmental delay risk according to EDI categories.

The model was fitted using a proportional odds logistic regression (POLR) via the polr function from the MASS package. The explanatory model included childhood obesity, physical activity compliance, maternal knowledge of complementary feeding, child's sex, child's age group, and anemia, the latter incorporated based on the causal framework specified by the Directed Acyclic Graph (DAG). For the development of the nomogram, variable selection emphasized parsimony and clinical interpretability, retaining only predictors that were clinically relevant and statistically supported within the explanatory model.

Model performance was evaluated in the validation set by calculating overall accuracy and Cohen's kappa statistic. Predicted probabilities were additionally stored to assess the stability of risk estimates across resampling iterations.

Mean accuracy and Cohen's kappa values across the 1,000 repetitions were computed to summarize internal model consistency and robustness. These metrics were reported to characterize model stability and internal reproducibility, rather than to claim diagnostic or prognostic accuracy.

### Nomogram

To facilitate clinical interpretation and individualized probabilistic risk estimation, a nomogram was generated based on the final multivariable ordinal logistic regression model fitted to the full dataset using the lrm function from the rms package. The nomogram graphically represents the predicted probability of neurodevelopmental delay risk, operationalized according to the EDI categories, based on the combined effect of the selected predictor variables. Predicted logits were transformed into probabilities using the logistic function.

For practical clinical application, the nomogram displays risk estimates across two outcome levels—normal development versus developmental lag or risk of developmental delay—while preserving the ordinal structure of the underlying model.

Statistically significant (p < 0.05) and clinically relevant variables were selected from the final model, including childhood obesity, physical activity compliance, child's sex, child's age group, and maternal knowledge of complementary feeding. Model coefficients were converted into proportionally scaled points to construct the nomogram.

Internal validation was performed using bootstrap resampling with 1,000 iterations. Model performance was evaluated using the area under the receiver operating characteristic curve (AUC), overall accuracy, and Cohen's kappa statistic [42] to assess internal performance and stability. Despite this internal validation, the nomogram should be interpreted as a tool for probabilistic risk estimation derived from an explanatory model, rather than as a standalone predictive or diagnostic instrument.

## Results

### Population characteristics

A total of 442 children under five years of age were evaluated between May 2022 and November 2024. The sex distribution was 57% male and 43% female. The mean age of the children was 22.94 ± 18.43 months; 42% were under one year of age, 16% were between one and two years, and 42% were aged three to four years. The mean age of mothers or caregivers was 31.81 ± 6.51 years. Regarding educational attainment, 42% had completed higher education, 33% had technical education, 23% had secondary education, and fewer than 3% had only primary or incomplete higher education.

With respect to nutritional status, 27% of the children were classified as obese, and 19% had anemia. Ninety-five percent met the recommended physical activity guidelines. Based on the EDI test, 58% of the children had normal development, 36% showed developmental lag, and 6% were at risk of developmental delay. Regarding caregiver knowledge of complementary feeding, 50% had a low level, 42.6% a moderate level, and 7.4% a high level (Table 1).

### Population characteristics by EDI developmental category

When the population was stratified by developmental status (normal development, developmental lag, or risk of developmental delay), statistically significant differences were observed across several variables. Male sex was more prevalent among children with developmental lag (64%) and those at risk of developmental delay (73%) compared with children with normal development (51%) (p = 0.010). Children with developmental lag or at risk of developmental delay had higher mean ages (28.28 ± 19.01 and 33.00 ± 17.56 months, respectively) than those with normal development (18.42 ± 16.60 months) (p < 0.001). Among children with normal development, 52% were younger than one year of age, whereas only 12% of children at risk of developmental delay belonged to this age group (p < 0.001).

Obesity prevalence was significantly higher among children with developmental lag (38%) and those at risk of developmental delay (46%) compared with children with normal development (17%) (p < 0.001). Likewise, non-compliance with physical activity recommendations was more frequent in the developmental lag group (7.1%) and in children at risk of developmental delay (31%) than in the normal development group (0.4%) (p < 0.001). Mothers of children with developmental lag or at risk of developmental delay also demonstrated lower levels of complementary feeding knowledge, with low knowledge observed in 68% and 69% of these groups, respectively, compared with 37% in the normal development group (p < 0.001).

No statistically significant differences were observed between groups with respect to anemia status (p = 0.14) or maternal age (p = 0.5). Maternal educational level differed across neurodevelopmental categories (p = 0.015). Mothers of children at risk of developmental delay were the only group in which incomplete higher education was observed (3.8%), whereas this category was rare among mothers of children with normal development (1.2%) and absent in the

**Table 1. Demographic and Clinical Characteristics of Children Under 5 Years of Age and Their Mothers/Caregivers Evaluated at Carlos Alberto Cortez Jiménez Hospital (EsSalud) and Zarumilla Health Center, Tumbes, 2024 (N = 442).**

| Characteristic | N | % |
|---|---|---|
| **Childs sex** | | |
| Male | 254 | 57 |
| Female | 188 | 43 |
| **Age (months)** (Mean ± SD) | | 22.94 ± 18.43 |
| **Age Group** | | |
| < 1 year | 184 | 42 |
| 1 to 2 years | 73 | 16 |
| 3 to 4 years | 185 | 42 |
| **Mother's Age (years)** (Mean ± SD) | | 31.81 ± 6.51 |
| **Maternal education level** | | |
| Primary education | 5 | 1.1 |
| Secondary education | 105 | 23 |
| Technical higher education | 144 | 33 |
| Incomplete higher education | 4 | 0.9 |
| Completed higher education | 184 | 42 |
| **Childhood obesity** [a] | | |
| Yes | 119 | 27 |
| No | 323 | 73 |
| **Anemia**[b] | | |
| Yes | 83 | 19 |
| No | 349 | 81 |
| Missing | 10 | – |
| **Physical Activity Compliance**[c] | | |
| Does not comply | 22 | 5 |
| Complies | 420 | 95 |
| **Neurodevelopmental delay risk (EDITest Result)** | | |
| Normal | 256 | 58 |
| Developmental lag | 159 | 36 |
| Risk of developmental delay | 27 | 6 |
| **Maternal knowledge of complementary feeding** | | |
| Low | 221 | 50 |
| Medium | 188 | 42.6 |
| HIgh | 32 | 7.4 |
| Missing | 1 | – |

[a]BMI-for-age z-score > +2 SD (WHO standards).

[b]Age-specific hemoglobin thresholds were used to define anemia (<13.5 g/dL for <2 months, <9.5 g/dL for 2–5 months, and <11.0 g/dL for 6–59 months) (35).

[c]< 1 year: active play several times per day, including ≥30 minutes of supervised prone positioning; 1–2 years: ≥ 180 minutes/day of physical activity of any intensity; 3–4 years: ≥ 120 minutes/day, including ≥60 minutes of moderate-to-vigorous activity. Outcome: meets or does not meet (WHO recommendations).

developmental lag group (0%). In contrast, the proportion of mothers with completed university education was similar across the normal, developmental lag, and at-risk groups (41%, 43%, and 46%, respectively) (Table 2).

## Multivariate ordinal regression analysis

An ordinal proportional odds regression model was applied to identify variables independently associated with neurodevelopmental delay risk in children, as assessed by the EDI test (Table 3). Female sex was associated with lower cumulative

**Table 2. Demographic and Clinical Characteristics of Children Under 5 Years of Age and Their Mothers/Caregivers According to EDI Test Results at Carlos Alberto Cortez Jiménez Hospital (EsSalud) and Zarumilla Health Center, Tumbes, 2024 (N = 431*).**

| Variable | Normal[1] (N = 250) | Developmental lag[1] (N = 155) | Risk of developmental delay[1] (N = 26) | p-value[2] |
|---|---|---|---|---|
| **Sex** | | | | 0.010 |
| Male | 128/ 250 (51%) | 99/ 155 (64%) | 19/ 26 (73%) | |
| Female | 122/ 250 (49%) | 56/ 155 (36%) | 7/ 26 (27%) | |
| **Age Group** | | | | <0.001 |
| <1 year old | 130/ 250 (52%) | 46/ 155 (30%) | 3/ 26 (12%) | |
| 1–2 years old | 49/ 250 (20%) | 19/ 155 (12%) | 5/ 26 (19%) | |
| 3–4 years old | 71/ 250 (28%) | 90/ 155 (58%) | 18/ 26 (69%) | |
| **Age (months)** (Mean ± SD) | 18.42 (16.60) | 28.28 (19.01) | 33.00 (17.56) | <0.001 |
| **Childhood obesity[a]** | | | | <0.001 |
| No | 208/ 250 (83%) | 96/ 155 (62%) | 14/ 26 (54%) | |
| Yes | 42/ 250 (17%) | 59/ 155 (38%) | 12/ 26 (46%) | |
| **Anemia[b]** | | | | 0.14 |
| No | 194/ 250 (78%) | 131/ 155 (85%) | 23/ 26 (88%) | |
| Yes | 56/ 250 (22%) | 24/ 155 (15%) | 3/ 26 (12%) | |
| **Physical Activity Compliance[c]** | | | | <0.001 |
| Does not comply | 1/ 250 (0.4%) | 11/ 155 (7.1%) | 8/ 26 (31%) | |
| Complies | 249/ 250 (99.6%) | 144/ 155 (92.9%) | 18/ 26 (69%) | |
| **Maternal knowledge of complementary feeding** | | | | <0.001 |
| Low | 92/ 250 (37%) | 105/ 155 (68%) | 18/ 26 (69%) | |
| Medium | 132/ 250 (53%) | 44/ 155 (28%) | 8/ 26 (31%) | |
| High | 26/ 250 (10%) | 6/ 155 (3.9%) | 0/ 26 (0%) | |
| **Mother's Age (years)** (Mean ± SD) | 31.56 (6.09) | 31.83 (6.75) | 34.04 (8.89) | 0.5 |
| **Maternal education level** | | | | 0.015 |
| Primary | 2/ 250 (0.8%) | 2/ 155 (1.3%) | 0/ 26 (0%) | |
| Secondary | 51/ 250 (20%) | 45/ 155 (29%) | 7/ 26 (27%) | |
| Technical | 92/ 250 (37%) | 42/ 155 (27%) | 6/ 26 (23%) | |
| Incomplete higher education | 3/ 250 (1.2%) | 0/ 155 (0%) | 1/ 26 (3.8%) | |
| University (completed) | 102/ 250 (41%) | 66/ 155 (43%) | 12/ 26 (46%) | |

[1] Mean (Standard Deviation); **n/ N (%)**

[2] Kruskal–Wallis rank sum test; Pearson's Chi-squared test; Fisher's exact test

[a]BMI-for-age z-score > +2 SD (WHO standards).

[b]Age-specific hemoglobin thresholds were used to define anemia (<13.5 g/dL for <2 months, <9.5 g/dL for 2–5 months, and <11.0 g/dL for 6–59 months) (35)..

[c]<1 year: active play several times per day, including ≥30 minutes of supervised prone positioning; 1–2 years: ≥ 180 minutes/day of physical activity of any intensity; 3–4 years: ≥ 120 minutes/day, including ≥60 minutes of moderate-to-vigorous activity. Outcome: meets or does not meet (WHO recommendations).

* Analyses were performed using complete cases only (n = 431). Participants with missing data in any key variable were excluded from this analysis.

**Table 3. Results of the Proportional Ordinal Logistic Regression Model for Child Development Level Among Children Under 5 Years and Their Mothers/Caregivers According to the EDI Test at Carlos Alberto Cortez Jiménez Hospital (EsSalud) and Zarumilla Health Center, Tumbes, 2024 (N = 431*).**

| Variable | Odds Ratio | Low CI 95% | High CI 95% | p-value |
|---|---|---|---|---|
| **Sex** | | | | |
| Male | 1.64 | 1.06 | 2.56 | 0.027 |
| **Age Group** | | | | |
| 1–2 years | 1.93 | 0.94 | 3.96 | 0.07 |
| 3–4 years | 7.89 | 2.40 | 25.91 | <0.001 |
| **Age (months)** | 0.98 | 0.95 | 1.01 | 0.19 |
| **Childhood Obesity[a] (yes)** | 2.73 | 1.66 | 4.51 | <0.001 |
| **Anemia (yes)[b]** | 0.80 | 0.44 | 1.45 | 0.46 |
| **Physical Activity Compliance (no)[c]** | 1.09 | 0.42 | 2.83 | 0.87 |
| **Dietary Knowledge** | | | | |
| Medium | 0.32 | 0.20 | 0.50 | <0.001 |
| High | 0.17 | 0.06 | 0.46 | <0.001 |
| **Maternal Age (years)** | 0.98 | 0.95 | 1.01 | 0.28 |
| **Educational Level** | | | | |
| Secondary | 0.71 | 0.08 | 6.60 | 0.76 |
| Technical | 0.32 | 0.03 | 3.00 | 0.32 |
| Incomplete University | 0.08 | 0.00 | 1.99 | 0.12 |
| Complete University | 0.57 | 0.06 | 5.30 | 0.62 |

**Note:** Odds ratios (ORs) were obtained from a proportional odds ordinal logistic regression model. ORs greater than 1 indicate increased cumulative odds of belonging to a higher (more impaired) developmental category, while ORs less than 1 indicate decreased cumulative odds (protective effect). Associations with $p < 0.05$ were considered statistically significant.

[a] BMI-for-age z-score > +2 SD (WHO standards).

[b] Age-specific hemoglobin thresholds were used to define anemia (<13.5 g/dL for <2 months, <9.5 g/dL for 2–5 months, and <11.0 g/dL for 6–59 months) (35).

[c] <1 year: active play several times per day, including ≥30 minutes of supervised prone positioning; 1–2 years: ≥180 minutes/day of physical activity of any intensity; 3–4 years: ≥120 minutes/day, including ≥60 minutes of moderate-to-vigorous activity. Outcome: meets or does not meet (WHO recommendations

* Analyses were performed using complete cases only (n = 431). Participants with missing data in any key variable were excluded from this analysis.

odds of belonging to a more impaired neurodevelopmental category (OR = 0.61; 95% CI: 0.39–0.94; p = 0.027), indicating a higher risk among male children.

Regarding age group, children aged 3–4 years showed significantly higher cumulative odds of neurodevelopmental delay risk compared with those younger than one year (OR = 7.89; 95% CI: 2.40–25.91; p < 0.001). The association for children aged 1–2 years did not reach statistical significance (OR = 1.93; 95% CI: 0.94–3.96; p = 0.07).

Childhood obesity was strongly associated with increased neurodevelopmental delay risk (OR = 2.73; 95% CI: 1.66–4.51; p < 0.001).

Higher maternal knowledge of complementary feeding showed a clear protective association. Compared with low knowledge, medium knowledge was associated with lower cumulative odds of developmental impairment (OR = 0.32; 95% CI: 0.20–0.50; p < 0.001), while high knowledge demonstrated an even stronger protective effect (OR = 0.17; 95% CI: 0.06–0.46; p < 0.001).

Child age in months, anemia status, physical activity compliance, maternal age, and maternal educational level were not significantly associated with neurodevelopmental delay risk in the multivariable model (Table 3).

## Model training validation and nomogram

To assess the classification potential of the model, we conducted 1,000 simulation runs. The results demonstrated robust performance, with accuracy indices ranging from 54.2% to 78.8% (median: 66.49%), the Cohen's Kappa values ranging from 4.4% to 56.59% (median: 31.93%), and the area under the curve (AUC) for discriminating each of EDI categories was of 0.76 (EDI1), 0.71(EDI2), and 0.74 (EDI3).

Additionally, using the previously selected predictive variables, we developed a nomogram to visualize and estimate individual probabilities of developmental outcomes (Fig 1).

Based on the clinical nomogram, a digital predictive calculator was developed using the Shiny platform (RStudio®), which is available openly at:

https://omicsperulab.shinyapps.io/NeuroDev_Calculator/

This tool allows for individualized estimation of the risk of neurodevelopmental disorders in children under five years of age, integrating statistically significant variables from the multivariate model (childhood obesity, age, sex, physical activity compliance, and maternal knowledge of complementary feeding). Each combination of factors generates a total score that translates into a personalized probability, facilitating its application in clinical and community settings.

The calculator offers a user-friendly visualization and immediate results, allowing its application in clinical and community settings for the early detection of neuropsychological risk in childhood.

## Discussion

Childhood obesity emerged as a frequent condition in the studied population, reinforcing concerns about the growing burden of excess weight in early childhood in Latin America [44]. The high occurrence observed in this setting aligns with regional reports indicating a shift in the nutritional profile of young children, particularly in low- and middle-income countries undergoing rapid epidemiological and nutritional transitions [45]. These findings highlight the relevance of childhood obesity as a contemporary public health challenge with potential implications for early neurodevelopment [46].

In parallel, anemia remained a common condition among children under five years of age in this cohort. Although its prevalence was lower than national estimates reported in Peru [47], anemia continues to represent a clinically meaningful

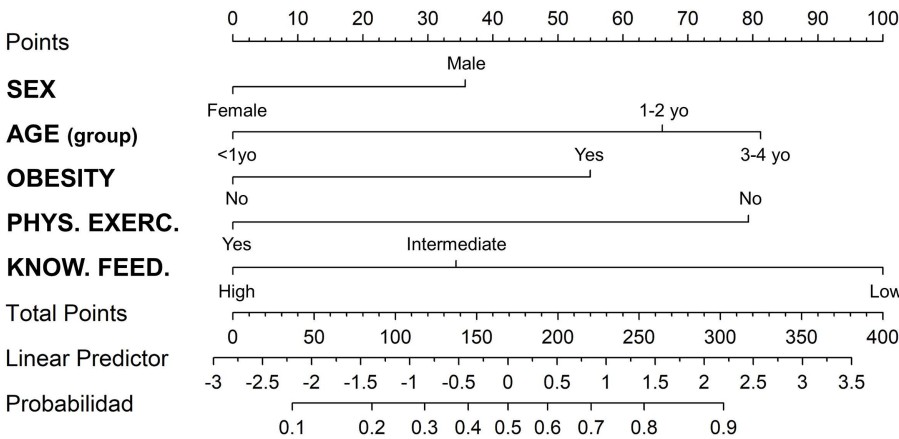

**Fig 1. Nomogram for Predicting Risk of Developmental Delay in Children Under 5 Years of Age.** Variables included are child sex, age group, presence of obesity, physical activity compliance, and Maternal Knowledge of complementary feeding. Each predictor contributes a number of points (top axis), which are summed to yield a total score (Total Points axis). This total score should be translated into a linear predictor value, which corresponds to the probability of the child being classified in a higher-risk developmental category. PHYS. EXERC.: PHYSICAL EXERCISE; KNOW. FEED.: KNOWLEDGE OF FEEDING.

concern given its well-established associations with adverse neurodevelopmental outcomes. Evidence from previous studies has linked early-life anemia, particularly iron deficiency, to alterations in myelination, language acquisition, behavior, and attention [48].Within this context, the coexistence of nutritional disorders such as obesity and anemia underscores the complexity of early developmental vulnerability and the need for integrated approaches when evaluating neurodevelopmental risk.

A significant association was observed between childhood obesity and neurodevelopmental delay risk in the present study. This finding aligns with prior evidence indicating that obesity in childhood is frequently accompanied by neurodevelopmental vulnerabilities. For example, a study conducted in a pediatric obesity clinic in Sweden reported a high prevalence of clinically diagnosed neurodevelopmental conditions—predominantly attention deficit hyperactivity disorder (ADHD)—among children with obesity and documented lower full-scale IQ and working memory scores among those with neurodevelopmental diagnoses [8]. Although that study was conducted in a specialized clinical setting and relied on diagnostic assessments rather than screening, both studies underscore a potential coexistence between obesity and neurodevelopmental vulnerability.

Evidence also suggests that obesity may be associated with differences in early motor development. Studies using developmental screening instruments such as the Ages and Stages Questionnaire (ASQ) and the Denver II test have reported disparities in gross motor skills—including jumping, running, and balance—between children with obesity and those with normal weight [49]. Similarly, Canadian studies in children under five with severe obesity have described a high prevalence of global developmental delay, particularly in the language domain, along with multiple clinical and psychosocial comorbidities in this population [50].

Taken together, these findings provide contextual support for the association observed in our study between childhood obesity and a higher risk of neurodevelopmental delay as identified through the EDI screening tool.

Evidence from interventional studies provides relevant background on the role of physical activity in early neurodevelopment. A recent meta-analysis reported that structured physical exercise interventions in children were associated with improvements in general and fluid intelligence compared with control groups [51]. The lack of an independent association between physical activity compliance and neurodevelopmental delay risk in the present study should be interpreted in light of methodological considerations. Physical activity was operationalized as a dichotomous, caregiver-reported indicator of compliance with age-specific recommendations, which may have limited variability and reduced sensitivity to detect graded associations. In addition, the very high observed compliance rate suggests a potential ceiling effect and the influence of social desirability bias, which may have attenuated observed associations.

In our study, male sex was significantly associated with a higher likelihood of neurodevelopmental delay risk compared to females.

This finding is consistent with emerging literature describing phenotypic, biological, and social sex- and gender-related differences in neurodevelopmental outcomes. Previous studies have suggested that factors such as prenatal exposure to sex hormones, X-linked gene regulation, and sex-related differences in brain organization have been proposed as potential mechanisms underlying observed sex differences in neurodevelopment. Additionally, gender-related socialization patterns from early childhood have been described as associated with differences in behavioral profiles, which may contribute to variation in developmental trajectories and patterns of identification between boys and girls [52].

We also observed a trend indicating a higher likelihood of neurodevelopmental delay risk with increasing age in months.

This finding may reflect the progressive and dynamic nature of early neurodevelopment, which involves age-dependent biological and functional maturation processes. Previous literature has described that brain maturation follows distinct trajectories in gray and white matter, with progressive hemispheric specialization. Language and motor functions are primarily lateralized to the left hemisphere, whereas visuospatial and attentional functions are more strongly associated with right hemispheric networks. These latter functions rely on more complex neural circuits that mature later in development and have been described as potentially more susceptible to developmental variation [53].

Our results also indicate that higher maternal knowledge of complementary feeding was significantly associated with a lower likelihood of neurodevelopmental delay risk. This finding is consistent with growing evidence describing associations between diet quality, the early nutritional environment, and neurodevelopmental outcomes in childhood. For example, a recent meta-analysis reported that healthy dietary patterns—characterized by higher intake of fruits, vegetables, and fiber, and lower consumption of saturated fats and added sugars—were associated with a reduced risk of internalizing and externalizing mental health problems in children. Several biological pathways have been proposed in the literature to explain these associations, including mechanisms related to gut–brain axis signaling, oxidative stress, neuroinflammatory processes, and the role of gut microbiota in neurodevelopment. Within this context, adequate caregiver knowledge of complementary feeding may represent an important modifiable factor associated with early developmental vulnerability, particularly during the first 1,000 days of life, a critical period for brain development [54].

However, given the cross-sectional design of the study, the possibility of reverse causality must be explicitly considered. Lower levels of physical activity may be associated with a higher probability of neurodevelopmental delay; however, it is also plausible that children with early neurodevelopmental vulnerability present motor limitations, reduced coordination, or lower initiative for movement, which in turn may lead to decreased participation in physical activity and increased sedentary behavior, as described in previous studies [55]. Accordingly, physical activity should be interpreted as a correlational marker within a potentially bidirectional relationship with neurodevelopmental vulnerability, rather than as a unidirectional determinant.

Similarly, while obesity may be associated with an increased probability of neurodevelopmental delay through mechanisms proposed in the literature, it is also possible that children at risk of neurodevelopmental delay engage in lower levels of physical activity, exhibit higher sedentary behavior, or experience difficulties in behavioral self-regulation, which may increase the likelihood of obesity [56]. Therefore, the temporal direction of these associations cannot be established in the present study. All findings should be interpreted as associations, underscoring the need for prospective longitudinal studies to clarify these bidirectional relationships.

Given the identification of multiple variables statistically associated with neurodevelopmental delay risk—including obesity, physical inactivity, lower mother knowledge of complementary feeding, sex, and age—we developed a multivariable nomogram as a clinical support tool to facilitate individualized estimation of neurodevelopmental delay risk in children under five years of age.

The nomogram was constructed based on an ordinal logistic regression model incorporating childhood obesity, physical activity compliance, sex, age group, and maternal knowledge of complementary feeding selected for their statistical significance and clinical relevance [8,51–54]. This tool allows healthcare professionals to visually estimate an individual child's probability of presenting a higher risk of neurodevelopmental delay, thereby supporting risk stratification and decision-making in primary care settings.

To ensure coherence with the explanatory causal framework underlying the study, variable selection for the nomogram was guided by a directed acyclic graph. Accordingly, variables conceptualized as potential mediators—such as anemia—were intentionally not included in the model. Although inclusion of mediators may improve apparent predictive performance by incorporating downstream information, this approach was avoided to preserve causal interpretability and to prevent overadjustment. The nomogram was therefore designed as an explanation-informed, clinically interpretable decision-support tool rather than a purely data-driven predictive model optimized solely for maximal discrimination.

Although nomograms have been widely used in pediatric research [57–59], their application to neurodevelopmental risk prediction in early childhood remains limited, particularly in low- and middle-income countries. Previous studies have shown that predictive models relying on complex laboratory or immunological markers may demonstrate high discriminatory performance but are often impractical in low-resource or primary care contexts [26,60]. In contrast, nomograms based on accessible clinical and sociodemographic variables have demonstrated good performance and greater feasibility for broader implementation [26,61,62].

The potential applicability of the proposed nomogram extends beyond insured healthcare settings, especially in contexts where access to specialized neurodevelopmental assessment tools is limited. In underserved populations and healthcare systems outside formal insurance networks, early identification of neurodevelopmental risk frequently relies on simplified screening strategies integrated into primary care or community-based programs [63]. In such settings, nomograms combining readily available clinical, nutritional, and sociodemographic information may represent a pragmatic and scalable approach to early risk identification [26].

Nevertheless, the external validity of this nomogram should be interpreted with caution. Differences in socioeconomic conditions, nutritional profiles, healthcare access, and early stimulation environments across populations may influence both predictor distributions and their associations with neurodevelopmental outcomes. Methodological literature emphasizes that prediction models developed in specific healthcare contexts often require external validation and recalibration before application in populations with different socioeconomic or healthcare system characteristics [64]. Accordingly, future studies should evaluate the performance of this nomogram in diverse settings, including rural communities and uninsured populations, to determine its generalizability and potential need for contextual adaptation.

In practical terms, the nomogram should be considered primarily as a decision-support tool for risk stratification in primary care and community health settings, rather than as a diagnostic or universally generalizable predictive model.

Taken together, the findings of this study highlight the potential relevance of addressing childhood obesity within primary healthcare settings, with particular attention to healthy lifestyle factors such as nutrition and physical activity during early childhood. In this context, the integration of early neurodevelopmental screening into routine child growth and development visits, using validated tools such as the EDI, may support the timely identification of children at increased neurodevelopmental delay risk.

Educational approaches targeting parents and caregivers, focused on healthy eating, early stimulation, and obesity prevention, could be particularly relevant for vulnerable families. Similarly, the incorporation of age-appropriate physical activity within early stimulation programs may help support motor, cognitive, and social development. Finally, continued attention to nutritional status and anemia monitoring in children under five years of age may be especially pertinent in regions such as Tumbes, where an epidemiological transition from undernutrition to obesity has been observed.

## Limitations

An important limitation of this study relates to the representativeness of the sample, which was restricted to children receiving care within the EsSalud healthcare network. As EsSalud primarily serves formally employed populations and their dependents, the study sample may differ from children attending public health facilities under the Ministry of Health (MINSA) or those without regular access to healthcare services, potentially limiting external validity [65].

This restriction may have influenced the observed prevalence estimates of obesity, anemia, and neurodevelopmental delay risk, which should therefore be interpreted with caution [66]. With respect to effect measures, while selection into the EsSalud system may influence baseline risk, the internal validity of the observed associations is less likely to be substantially compromised, if exposure–outcome relationships operate similarly within this population.

Post-stratification weighting and sensitivity analyses to address differential healthcare access were considered; however, these approaches were not feasible due to the absence of population-level reference data for the EsSalud pediatric population and the cross-sectional nature of the study.

Regarding effect measures, while the internal associations between childhood obesity, physical activity compliance, caregiver knowledge of complementary feeding, and neurodevelopmental delay risk are likely valid within this healthcare-based sample, caution is warranted when extrapolating these findings to broader or socioeconomically distinct populations. No post-stratification weighting or sensitivity analyses were performed, as the study was not designed to be population-representative but rather to explore associations within a defined healthcare setting [67].

Physical activity was assessed using a caregiver-reported questionnaire based on World Health Organization recommendations rather than a formally validated physical activity scale. Although this approach allowed classification of adherence to international guidelines, it may be subject to recall bias and social desirability bias. In addition, dichotomization of physical activity may have resulted in a loss of information by not capturing gradations in activity intensity or duration [68]. Nevertheless, this operationalization was considered appropriate given the study's focus on compliance with established public health thresholds and the inherent limitations of caregiver-reported data in very young children. Methodological evidence suggests that, in the presence of measurement error or contamination, dichotomization may reduce noise and yield more robust estimates when clinically meaningful cut-off points are applied [69]. Future studies incorporating validated questionnaires or objective measures such as accelerometry may provide a more nuanced assessment of dose–response relationships.

The relatively high prevalence of compliance with physical activity recommendations observed in this study should also be interpreted with caution. Caregiver-reported data may be influenced by social desirability, leading to overreporting of behaviors perceived as healthy or socially acceptable [70]. Additionally, the dichotomous classification based on guideline thresholds may contribute to an apparent inflation of compliance rates, as it does not capture variability below or above recommended cut-offs duration [68]. The healthcare setting in which data were collected, together with caregivers' awareness of recommended behaviors, may have further increased the likelihood of socially desirable responses [71].

Similarly, the operationalization of anemia as a dichotomous variable across all age groups may have resulted in loss of information regarding anemia severity among older children. However, this approach allowed a uniform and analytically consistent definition of anemia across the entire study population, given that national guidelines do not permit severity classification in infants younger than six months.

Finally, because anemia was positioned as a potential mediator in the causal framework defined by the directed acyclic graph (obesity→anemia→neurodevelopmental delay risk), it was not included in the primary explanatory model to avoid overadjustment and attenuation of the total effect of obesity. Although inclusion of mediators may improve apparent predictive performance by incorporating downstream information, this comes at the cost of reduced causal interpretability; therefore, mediator inclusion would be more appropriate for a purely predictive objective rather than for explanatory analysis

## Conclusion

This study identified a significant association between childhood obesity and neurodevelopmental delay risk among children under five years of age receiving care within the EsSalud healthcare network in Tumbes, Peru. Children with obesity showed a higher likelihood of presenting developmental lag or being classified as at risk of developmental delay according to the EDI screening tool.

Male sex was independently associated with a higher likelihood of belonging to a more impaired neurodevelopmental category. Higher caregiver knowledge of complementary feeding was inversely associated with neurodevelopmental delay risk, highlighting the potential relevance of caregiver-related factors in early child development. In contrast, physical activity compliance was not independently associated with neurodevelopmental delay risk in the multivariable analysis. These findings should be interpreted within the context of the study's cross-sectional design, recognizing the possibility of bidirectional relationships and reverse causality, particularly with respect to physical activity.

Although maternal age and anemia were not statistically significant in the multivariable model, the observed prevalence of anemia remains clinically relevant and warrants further investigation, particularly given its potential role within the broader causal pathway influencing child development.

The nomogram developed in this study provides a structured approach for probabilistic estimation of neurodevelopmental delay risk in children under five years of age. Based on variables routinely available in primary care settings—such as childhood obesity status, child's age group, sex, physical activity compliance, and caregiver knowledge of complementary

feeding—the model demonstrated acceptable discriminatory performance in internal validation (AUC > 0.7). However, it should be interpreted as an explanation-informed decision-support tool rather than a definitive predictive instrument, and its applicability may vary according to clinical context.

The availability of this nomogram as an online digital calculator may support its integration into community and clinical settings to facilitate early risk identification and risk stratification, contributing to informed and individualized decision-making in child health.

## Supporting information

**S1 Material. Informed Consent to participate in this study (Spanish and English versions).**
(DOCX)

**S2 Material. Complete Directed Acyclic Graph of this study [72–79].**
(DOCX)

**S3 Material. STROBE Statement—checklist of items that should be included in reports of observational studies.**
(DOCX)

## Author contributions

**Conceptualization:** Miriam Arredondo-Nontol.

**Data curation:** Rodolfo Arredondo-Nontol, Narcisa Reto, Alexis Germán Murillo Carrasco.

**Formal analysis:** Rodolfo Arredondo-Nontol, Narcisa Reto, Alexis Germán Murillo Carrasco.

**Investigation:** Miriam Arredondo-Nontol.

**Methodology:** Miriam Arredondo-Nontol.

**Project administration:** Miriam Arredondo-Nontol.

**Supervision:** Miriam Arredondo-Nontol.

**Visualization:** Rodolfo Arredondo-Nontol, Narcisa Reto, Alexis Germán Murillo Carrasco.

**Writing – original draft:** Miriam Arredondo-Nontol.

**Writing – review & editing:** Rodolfo Arredondo-Nontol, Narcisa Reto, Alexis Germán Murillo Carrasco.

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
