## [Decision Letter · Decision Letter 0]

9 Jan 2026

Dear Dr. Reto,

Thank you for submitting your manuscript to PLOS ONE. After careful consideration, we feel that it has merit but does not fully meet PLOS ONE’s publication criteria as it currently stands. Therefore, we invite you to submit a revised version of the manuscript that addresses the points raised during the review process.

**ACADEMIC EDITOR:**

1) Conceptual aspects and terminology:

- Standardize the terminology related to the outcome, avoiding the interchangeable use of “neurodevelopmental delay” and “developmental disorders”.

- Revise the Discussion and Conclusion to avoid causal inferences, ensuring that the language reflects associations observed in a cross-sectional study.

- Include an explicit discussion of the possibility of reverse causality, particularly in the relationships between physical activity, obesity, and developmental delay.

2) Study design, sample, and ethical aspects:

- Clearly specify the inclusion and exclusion criteria of the sample.

- Indicate whether a sample size calculation was performed; if not, clarify whether the sample was defined as census-based or consecutively recruited.

- Justify the selection of the age group (<5 years) and clarify whether children with Autism Spectrum Disorder were included.

- Clarify whether the ethics approval covered any extensions of the data collection period.

- Expand the discussion on the limited representativeness of the sample (restricted to the EsSalud network), including potential impacts on prevalence estimates and effect measures, as well as whether post-stratification weighting or sensitivity analyses were performed.

3) Measurement and operationalization of variables

3.1) Physical activity:

- Provide a detailed description of the data collection instrument (questionnaire and specific questions used).

- Justify the dichotomization of the variable and discuss the associated loss of information.

- Explicitly describe how WHO recommendations were operationalized across different age groups,  particularly among children under one year of age.

- Critically discuss the high prevalence of compliance observed and the potential for social desirability bias.

3.2) Anemia:

- Specify the data source (laboratory measurements or caregiver report).

- Describe the cutoff values used, any altitude adjustments (if applicable), and the references adopted.

3.3) Maternal knowledge of complementary feeding:

- Detail the scoring system and cutoff points used.

- Create specific subsections in the Methods for physical activity and anemia to improve clarity and reproducibility.

4) Missing data and sociodemographic variables:

- Report the presence and proportion of missing data.

- Describe how missing data were handled in the analysis.

- Systematically present all sociodemographic variables collected and ensure consistency between the Methods, Results, and Tables.

5) Statistical analysis and modeling:

- Clarify the analytical objective of the study (explanatory vs. predictive) and align methodological decisions accordingly.

- Justify the exclusion of anemia based on the DAG in the context of a predictive model, or consider alternative analytical approaches.

- Discuss the impact of excluding potential mediators on model performance.

- Revisit the interpretation of the role of physical activity, explicitly acknowledging the possibility of a bidirectional association with the outcome.

6) Results, tables, and presentation:

- Ensure that all tables cited in the text are included and correctly numbered.

- Make tables self-explanatory by including cutoff values for obesity, anemia, and physical activity in table legends or notes.

- Correct inconsistencies and typographical errors (e.g., missing spaces before citations).

7) Discussion and applicability of the nomogram:

- Revise the opening paragraph of the Discussion to avoid repetition of the study objective and excessive description of results.

- Expand the discussion on the contexts in which the nomogram may be applied and its external validity, particularly in populations outside insured healthcare networks and across different socioeconomic settings.

8) References: Review and standardize all references, ensuring completeness, accuracy, and correct correspondence between in-text citations and the reference list.

We look forward to receiving your revised manuscript.

Kind regards,

Elma Izze Da Silva Magalhães

Academic Editor

PLOS One

**Journal Requirements:**

“This study was partially funded by the Institute for Health Technology Assessment and Research (IETSI), ESSALUD, through its Mentoring Program”

4. Please note that funding information should not appear in any section or other areas of your manuscript. We will only publish funding information present in the Funding Statement section of the online submission form. Please remove any funding-related text from the manuscript.

Reviewers' comments:

Reviewer's Responses to Questions

**Comments to the Author**

1. Is the manuscript technically sound, and do the data support the conclusions?

Reviewer #1: Yes

Reviewer #2: Partly

Reviewer #3: Yes

2. Has the statistical analysis been performed appropriately and rigorously?

Reviewer #1: Yes

Reviewer #2: N/A

Reviewer #3: Yes

3. Have the authors made all data underlying the findings in their manuscript fully available?

Reviewer #1: Yes

Reviewer #2: Yes

Reviewer #3: Yes

4. Is the manuscript presented in an intelligible fashion and written in standard English?

Reviewer #1: Yes

Reviewer #2: Yes

Reviewer #3: Yes

Reviewer #1: 1.Standardize the terminology.At times, the terms “neurodevelopmental delay” and “developmental disorders” are used interchangeably. Consider standardizing terminology to improve clarity.

2.Ensure all references are complete and accurate and cited in parentheses (e.g., (8), (46–49)) corresponding accurately to a complete reference list, particularly those supporting prevalence figures and previous nomogram applications.

3.Correct minor typographical issues (e.g., missing spaces before reference citations).

4.Check table numbering and ensure all tables referenced in the text are included in the submission.

5.Clarify whether ethics approval covered any extensions to the study period.

6.Provide operational details for physical activity compliance, anemia diagnosis (including any altitude adjustments), and scoring thresholds for complementary feeding knowledge.

7.The sample was drawn entirely from insured patients (EsSalud network), which may limit applicability to uninsured or rural populations. This limitation is mentioned briefly, but more detail is needed on how this might affect both prevalence estimates and effect measures, and whether any post-stratification weighting or sensitivity analyses were considered.

8.Although the manuscript acknowledges its cross-sectional design, parts of the discussion use language suggestive of causality (e.g., stating that obesity “increases” the risk of developmental delay). The authors should revise the text to reflect associations rather than causal effects. Additionally, the potential for reverse causation—such as developmental delay leading to reduced physical activity and subsequent obesity—should be explicitly addressed.

9.Any specific reason to select the sample size under five years of age ,as nothing is mentioned in manuscript.Children above 5 are also at high risk of obesity due to the spreading epidemic related to sedentary life style and poor dietrery habits.

10 Were children with Autism included in study?

Reviewer #2: The manuscript addresses a topic of significant public health interest: the association between childhood obesity and neurodevelopment. The authors are to be acknowledged for selecting this subject, particularly within the challenging context of the "double burden" of malnutrition. Furthermore, the study’s inclusion of physical activity as a focal point is noteworthy, as it appropriately draws attention to the role of movement in early childhood development beyond its function in weight management. However, despite these conceptual strengths, a detailed review of the manuscript reveals significant methodological limitations that currently compromise the validity of the findings. Specific concerns regarding study design, variable measurement, and statistical analysis are detailed below.

First, a fundamental methodological conflict exists between the study's dual goals of explaining an association and building a predictive tool. The authors employed a DAG to exclude "Anemia" on the basis that it acts as a "mediator". While excluding mediators is standard practice for establishing the isolated causal effect of obesity, it is often counterproductive for a predictive model. In prediction scenarios, mediators are frequently the strongest predictors and should be retained to maximize model accuracy. Consequently, the exclusion of anemia likely contributes to the model's suboptimal performance.

Regarding the PA indicator specifically, although the authors attempt to enhance the model by incorporating it as a key variable, there are substantial weaknesses concerning its measurement, distributional characteristics, and causal interpretation. The primary issue lies in the dichotomization of the variable. The study treats physical activity as binary—"Complies" vs. "Does not comply" with WHO guidelines —which results in a significant loss of information. By treating a child who barely meets the minimum threshold identically to a highly active child, the simple "Yes/No" classification fails to capture the gradient of intensity and movement type that is crucial in early childhood research and likely correlates with neurodevelopmental nuances.

This lack of granularity appears to have resulted in statistically anomalous data. The reported PA compliance rate is suspiciously high (95%) , which presents an epidemiological paradox when contrasted with the study's high obesity prevalence of 27%. Finding that nearly all children meet activity guidelines while nearly one-third are obese is contradictory, as low physical activity is typically a primary driver of obesity. This discrepancy strongly suggests that the measurement tool is either highly inaccurate or subject to significant social desirability bias.

Moreover, the validity of this assessment is further compromised by the age distribution of the sample. The evaluation of physical activity in children under one year old, who constitute 42% of the sample, is questionable. For an infant (<1 year), WHO guidelines focus on passive metrics like "tummy time" and "not being restrained," whereas for a 4-year-old, the guidelines involve energetic play and running. These are fundamentally different constructs. Asking a mother if a 6-month-old "meets physical activity guidelines" is cognitively more complex and prone to misinterpretation than reporting on a 4-year-old's running habits. The manuscript does not clarify how these disparate developmental milestones were standardized into a single variable.

Finally, there is a significant risk of reverse causality. The study identifies physical activity as a "protective factor", yet the relationship is likely bidirectional. The outcome variable (EDI test) explicitly measures gross motor skills. Children with developmental delays—the study's outcome of interest—may physically be unable to meet activity guidelines due to motor limitations such as hypotonia or coordination issues. Indeed, Table 2 reveals that in the "At Risk" group, non-compliance jumps to 30% compared to only 0.8% in the normal group. It is highly probable that their "inactivity" is a result of their developmental delay, rather than solely a cause.

In conclusion, while the initiative to create a practical tool for the early detection of developmental delays is commendable, the current operationalization of key variables significantly undermines the model's robustness. Specifically, the treatment of physical activity and the resulting high compliance rate suggest a disconnect between the data and the clinical reality of the population. Unless these methodological biases are rigorously corrected or re-analyzed, the conclusions regarding the protective factors for neurodevelopment remain open to question.

Reviewer #3: Congratulations on the study. It addresses a very important topic and is developed in a very consistent way. Please find my comments below:

Methods:

Was a sample size calculation performed? If so, please explain it. If not, clarify how the sample was defined. It appears that all children attending the growth and development units were included; please specify this clearly.

In addition, although you clearly explained the exclusion criteria in this section, what were the inclusion criteria? Please make these explicit.

Regarding physical activity adherence, how were the data collected? Was a questionnaire used? Which specific questions were included? Please specify.

Regarding anemia assessment, did you perform laboratory analyses to obtain this information, or was it reported by caregivers? Furthermore, please describe the cutoff values used to define anemia and cite the corresponding reference (as you did for obesity assessment). Please specify.

I suggest creating separate sections for physical activity adherence and anemia assessment in order to allow clearer and more detailed description of these aspects.

Did you have missing data? Please specify. If so, how were these data handled?

In the section “Maternal Knowledge of Complementary Feeding” you state: “Additional demographic variables were collected, including maternal age and educational attainment.” Please describe all demographic variables collected in the study and how they were obtained.

Also include the variables reported in the first section of the Results and in Tables 1 and 2, clarifying that they were assessed in the Methods section and explaining how they were measured.

Results:

In the descriptions of Tables 1, 2, and 3, please present the cutoff values used to define obesity, anemia, and physical activity. The tables should be interpretable without the need to refer back to the Methods section.

Discussion:

In the first paragraph, it is not necessary to restate the study objective. The rest of the paragraph describes results rather than providing discussion. Please review this.

In which contexts do you consider the use of the nomogram applicable? Please expand the discussion regarding the external validity of its use.

**Do you want your identity to be public for this peer review?** For information about this choice, including consent withdrawal, please see our Privacy Policy

Reviewer #1: **Yes:** Dr Muzna Arif

Reviewer #2: **Yes:** Guo Ye

Reviewer #3: **Yes:** Dafne Pavão Schattschneider

---

## [Author Response · Author response to Decision Letter 1]

9 Feb 2026

Response to Reviewers’ Comments

Association Between Obesity and Neurodevelopmental Disorders in Young Children: A Study from Tumbes, Peru

PONE-D-25-30834

Plos one

A. ACADEMIC EDITOR

Comment 1.1

Standardize the terminology related to the outcome, avoiding the interchangeable use of “neurodevelopmental delay” and “developmental disorders”.

Response 1.1

We agree with this observation. We have standardized the terminology related to the outcome throughout the manuscript, avoiding the interchangeable use of terms. Specifically, we consistently use the term “risk of neurodevelopmental delay” to refer to the outcome assessed through a developmental screening instrument, clearly distinguishing it from clinically diagnosed neurodevelopmental disorders.

Additionally, to ensure full conceptual coherence, we have revised the manuscript title to more accurately reflect that the study evaluates the risk of neurodevelopmental delay identified through screening, rather than formally diagnosed neurodevelopmental disorders. All sections of the manuscript (Abstract, Methods, Results, Discussion, tables, and figures) were reviewed and updated accordingly to align with this change.

Comment 1.2

Revise the Discussion and Conclusion to avoid causal inferences, ensuring that the language reflects associations observed in a cross-sectional study.

Response 1.2

The Discussion and Conclusion sections were comprehensively revised to ensure full coherence with the study’s cross-sectional design. All causal or prescriptive language was removed or reformulated using associative terminology (e.g., “was associated with,” “showed a higher probability,” “may reflect,” “has been described”). Interpretations were reframed to emphasize associations rather than causal relationships. In addition, the Conclusion was modified to highlight the observational nature of the findings, the probabilistic use of the nomogram, and the need for external validation prior to broader application.

Comment 1.3

Include an explicit discussion of the possibility of reverse causality, particularly in the relationships between physical activity, obesity, and developmental delay.

Response 1.3

We thank the reviewer for this important observation. In response, we have explicitly incorporated into the Discussion section a reflection on the possibility of reverse causality in the observed associations between physical activity, obesity, and the risk of neurodevelopmental delay. These relationships are acknowledged to be potentially bidirectional, as lower levels of physical activity and obesity may be associated with neurodevelopmental delay, while neurodevelopmental delay itself may also lead to reduced physical activity levels and an increased risk of obesity. Furthermore, we emphasize that, due to the cross-sectional design of the study, it is not possible to establish the temporal directionality of these associations.

Comment 1.4

Clearly specify the inclusion and exclusion criteria of the sample.

Response 1.4

We thank the reviewer for this observation. In response, the Study Design and Population section was revised to clearly specify the inclusion and exclusion criteria of the sample, which are now presented under a dedicated subheading.

Comment 1.5

Indicate whether a sample size calculation was performed; if not, clarify whether the sample was defined as census-based or consecutively recruited.

Response 1.5

We thank the reviewer for this observation. A sample size calculation was performed during the study planning phase based on the expected prevalence of the outcome of interest. However, the final sample size was determined by consecutive recruitment of all eligible children attending the participating healthcare facilities during the study period.

During the conduct of the study, participants were included through consecutive recruitment, enrolling all eligible children under five years of age who attended the participating facilities and whose parents or legal guardians provided informed consent. This clarification has been incorporated into the Study Design and Population section of the manuscript.

Comment 1.6

Justify the selection of the age group (<5 years) and clarify whether children with Autism Spectrum Disorder were included.

Response 1.6

We thank the reviewer for this observation. The selection of children under five years of age is justified by the fact that early childhood represents a critical period of neurodevelopment, highly sensitive to nutritional, metabolic, and behavioral factors, and also corresponds to the age range for which the Evaluación del Desarrollo Infantil (EDI) instrument has been validated. This justification has been incorporated into the Methods section.

In addition, the exclusion criteria have been clarified to specify that children with prior clinical diagnoses of neurodevelopmental disorders, including Autism Spectrum Disorder, were not included in the study.

Comment 1.7

Clarify whether the ethics approval covered any extensions of the data collection period.

Response 1.7

We thank the reviewer for this observation. The study received initial ethical approval in August 2021 from the Institutional Research Ethics Committee of the EsSalud Tumbes Healthcare Network. Ethical approval was subsequently renewed on an annual basis through official certifications issued in 2022, 2023, and 2024, thereby ensuring ethical oversight throughout the entire data collection period. This clarification has been incorporated into the Methods section of the manuscript.

Comment 1.8

Expand the discussion on the limited representativeness of the sample (restricted to the EsSalud network), including potential impacts on prevalence estimates and effect measures, as well as whether post-stratification weighting or sensitivity analyses were performed.

Response 1.8

We thank the reviewer for this comment. The Discussion section has been expanded to explicitly address the limited representativeness of the study sample, which was restricted to children receiving care within the EsSalud healthcare network. As EsSalud primarily serves formally employed populations and their dependents, the study population may differ from uninsured children, those attending Ministry of Health facilities, or populations in more rural or socioeconomically disadvantaged settings.

This restriction may primarily affect prevalence estimates, which should therefore be interpreted with caution, as children with regular access to healthcare services may differ in nutritional status, health-seeking behavior, and early detection of developmental risk. With respect to effect measures, while selection into the EsSalud system may influence baseline risk, the internal validity of the observed associations is less likely to be substantially compromised, assuming that exposure–outcome relationships operate similarly within this healthcare setting.

Post-stratification weighting and sensitivity analyses were considered; however, these approaches were not performed due to the absence of population-level reference distributions for the EsSalud pediatric population and the cross-sectional design of the study. These considerations are now explicitly stated in the Limitations section of the manuscript.

Comment 1.9

Provide a detailed description of the data collection instrument for physical activity (questionnaire and specific questions used).

Response 1.9

We thank the reviewer for this observation. The Methods section has been expanded to provide a more detailed description of the instrument used to assess physical activity. Physical activity was evaluated using a structured questionnaire administered to caregivers, specifically developed for this study and directly based on the World Health Organization’s physical activity recommendations for children under five years of age. The instrument classified compliance with these recommendations according to age groups and was used as a categorical variable in the analysis. In addition, it is acknowledged that the questionnaire does not constitute a formally validated scale, which has been identified as a limitation of the study.

Comment 1.10

Justify the dichotomization of the variable and discuss the associated loss of information.

Response 1.10

We thank the reviewer for this observation. The dichotomization of the physical activity variable was performed to classify compliance with the World Health Organization’s age-specific recommendations, prioritizing clinical interpretability and comparability across age groups in early childhood. In addition, a reflection on the potential loss of information associated with this methodological decision has been incorporated into the Discussion section, acknowledging that gradients of physical activity intensity or duration are not captured, which represents a limitation of the study.

Comment 1.11

Explicitly describe how WHO recommendations were operationalized across different age groups, particularly among children under one year of age.

Response 1.11

We thank the reviewer for this observation. The Methods section has been expanded to explicitly describe how the World Health Organization’s physical activity recommendations were operationalized according to age groups. In particular, the criterion applied to children under one year of age was detailed, based on engagement in active play and time spent in the prone position (“tummy time”) during the day, in accordance with WHO recommendations. In addition, the physical activity thresholds applied for the 1–2 and 3–4 year age groups were specified.

Comment 1.12

Critically discuss the high prevalence of compliance observed and the potential for social desirability bias.

Response 1.12

We thank the reviewer for this observation. The Discussion section has been expanded to critically address the high prevalence of compliance with physical activity recommendations observed in the study. In particular, we discuss the potential impact of social desirability bias associated with the use of caregiver-reported information, as well as the influence of dichotomous classification based on international guideline thresholds. These considerations have been incorporated as part of the study’s limitations.

Comment 1.13

Specify the data source for anemia (laboratory measurements or caregiver report).

Response 1.13

We thank the reviewer for this observation. The Methods section now specifies that anemia was determined exclusively through laboratory measurements, using hemoglobin concentration obtained from venous blood samples, and that no caregiver-reported information was used for its classification.

Comment 1.14

Describe the cutoff values used, any altitude adjustments (if applicable), and the references adopted.

Response 1.14

We thank the reviewer for this observation. The Methods section now specifies the cutoff points used for the definition of anemia, based on the Clinical Practice Guideline of the Peruvian Ministry of Health (MINSA). Specifically, anemia was defined as hemoglobin <13.5 g/dL in children younger than 2 months, <9.5 g/dL in children aged 2 to 5 months, and <11.0 g/dL in children aged 6 to 59 months.

In addition, it was clarified that, for analytical purposes, anemia was operationalized as a dichotomous variable (presence/absence) across all age groups, in order to ensure consistency and comparability in the analysis.

Given that the study was conducted in Tumbes, a coastal region located at sea level, no altitude adjustment was applied to hemoglobin values. All definitions and cutoff points used were based on the official recommendations of the Peruvian Ministry of Health.

Comment 1.15

Detail the scoring system and cutoff points used for maternal knowledge of complementary feeding.

Response 1.15

We thank the reviewer for this observation. The Methods section now details the scoring system used for the maternal knowledge questionnaire on complementary feeding. Each correct response was awarded one point, whereas incorrect or “do not know” responses received zero points. The total score was obtained by summing the correct responses and was used to classify maternal knowledge into categories (low, moderate, and good), which were subsequently used in the statistical analysis.

Comment 1.16

Create specific subsections in the Methods for physical activity and anemia.

Response 1.16

We thank the reviewer for the suggestion. In response, the Methods section was reorganized to incorporate specific subsections entitled “Physical Activity Assessment” and “Anemia Assessment,” in which the measurement procedures, operationalization, and criteria used for each variable are described in a detailed and separate manner, in order to improve the clarity and reproducibility of the study.

Comment 1.17

Report the presence and proportion of missing data.

Response 1.17

We thank the reviewer for this observation. Of the 445 initial records, 3 were excluded prior to analysis due to ineligibility (n = 2) or duplication (n = 1). Among the 442 eligible participants, 11 children (2.5%) had missing data in at least one key variable, corresponding to hemoglobin (n = 10) or the maternal questionnaire (n = 1), as detailed in Table 1.

Given the low proportion of missing data and the absence of systematic patterns, multivariable analyses were conducted using a complete-case approach, resulting in a final analytical sample of 431 children. No imputation methods were applied.

Comment 1.18

Describe how missing data were handled in the analysis.

Response 1.18

We thank the reviewer for this observation. Missing data were assessed prior to analysis. Records with incomplete information in key variables were excluded from the multivariable analysis, and a complete-case analysis was performed. No imputation methods were applied, as the proportion of missing data was low and no systematic patterns of missingness were identified.

Comment 1.19

Systematically present all sociodemographic variables collected and ensure consistency.

Response 1.19

We thank the reviewer for this observation. All sociodemographic variables collected (child age and sex, maternal age, and maternal educational level) are now presented in a systematic and consistent manner across the Methods and Results sections, as well as in the corresponding tables. In addition, the consistency of variable naming and categorization throughout the manuscript was verified, in accordance with recommendations for the reporting of observational studies.

Comment 1.20

Clarify the analytical objective (explanatory vs predictive).

Response 1.20

We thank the reviewer for this important comment. The primary analytical objective of the study is explanatory, aimed at estimating the association between childhood obesity and neurodevelopmental delay risk. Accordingly, we used an ordinal logistic regression model guided by a causal framework (DAG) to control for confounding and avoid overadjustment.

A nomogram and online calculator were developed as a secondary, applied objective to facilitate individual probabilistic risk estimation based on the explanatory model, rather than to optimize predictive performance. We have clarified this distinction in the Objectives and Statistical Analysis sections.

Comment 1.21

Justify the exclusion of anemia based on the DAG in the context of a predictive model, or consider alternative analytical approaches.

Response 1.21

We thank the reviewer for this insightful comment. In our study, anemia was included in the initial multivariable ordinal logistic regression model as part of the explanatory analysis, based on the causal assumptions defined in the Directed Acyclic Graph (DAG). This approach allowed us to evaluate its potential confounding role in the association between childhood obesity and neurodevelopmental delay risk.

However, anemia was not significantly associated with the outcome in the multivariable model and did not meaningfully contribute to model discrimination. Given that the primary analytical objective of the study was explanatory, and that the predictive component (nomogram and online calculator) was d

---

## [Editor Report · Decision Letter 1]

12 Feb 2026

Association Between Obesity and Neurodevelopmental Delay Risk in Children Under Five Years: A Study from Tumbes, Peru

PONE-D-25-30834R1

Dear Dr. Reto,

We’re pleased to inform you that your manuscript has been judged scientifically suitable for publication and will be formally accepted for publication once it meets all outstanding technical requirements.

Kind regards,

Elma Izze Da Silva Magalhães

Academic Editor

PLOS One

---

## [Editor Report · Acceptance letter]

PONE-D-25-30834R1

PLOS One

Dear Dr. Reto,

I'm pleased to inform you that your manuscript has been deemed suitable for publication in PLOS One. Congratulations! Your manuscript is now being handed over to our production team.

Kind regards,

on behalf of

Dr. Elma Izze Da Silva Magalhães

Academic Editor

PLOS One